# Does Humeral Component Version Affect Range of Motion and Clinical Outcomes in Reverse Total Shoulder Arthroplasty? A Systematic Review

**DOI:** 10.3390/jcm10245745

**Published:** 2021-12-08

**Authors:** Shivan S. Jassim, Lukas Ernstbrunner, Eugene T. Ek

**Affiliations:** 1Melbourne Orthopaedic Group, 33 The Avenue, Windsor, Melbourne, VIC 3181, Australia; shivan.jassim@doctors.org.uk (S.S.J.); lukas.ernstbrunner@icloud.com (L.E.); 2Department of Biomedical Engineering, University of Melbourne, Parkville, VIC 3010, Australia; 3Department of Orthopaedic Surgery, Royal Melbourne Hospital, Parkville, VIC 3050, Australia; 4Department of Surgery, Monash Medical Centre, Monash University, Melbourne, VIC 3181, Australia

**Keywords:** reverse total shoulder arthroplasty, implant design, humeral version, shoulder function, onlay, inlay

## Abstract

Background: Prosthesis selection, design, and placement in reverse total shoulder arthroplasty (RTSA) affect post-operative results. The aim of this systematic review was to evaluate the influence of the humeral stem version and prosthesis design (inlay vs. onlay) on shoulder function following RTSA. Methods: A systematic review of the literature on post-operative range of motion (ROM) and functional scores following RTSA with specifically known humeral stem implantations was performed using MEDLINE, Pubmed, and Embase databases, and the Cochrane Library. Functional scores included were Constant scores (CSs) and/or American Shoulder and Elbow Surgeons (ASES) scores. The patients were organised into three separate groups based on the implanted version of their humeral stem: (1) less than 20° of retroversion, (2) 20° of retroversion, and (3) greater than 20° of retroversion. Results: Data from 14 studies and a total of 1221 shoulders were eligible for analysis. Patients with a humeral stem implanted at 20° of retroversion had similar post-operative mean ASES (75.8 points) and absolute CS (68.1 points) compared to the group with humeral stems implanted at less than 20° of retroversion (76 points and 62.5 points; *p* = 0.956 and *p* = 0.153) and those implanted at more than 20° of retroversion (73.3 points; *p* = 0.682). Subjects with humeral stem retroversion at greater than 20° tended towards greater active forward elevation and external rotation compared with the group at 20° of retroversion (*p* = 0.462) and those with less than 20° of retroversion (*p* = 0.192). Patients with an onlay-type RTSA showed statistically significantly higher mean post-operative internal rotation compared to patients with inlay-type RTSA designs (*p* = 0.048). Other functional scores and forward elevation results favoured the onlay-types, but greater external rotation was seen in inlay-type RTSA designs (*p* = 0.382). Conclusions: Humeral stem implantation in RTSA at 20° of retroversion and greater appears to be associated with higher post-operative outcome scores and a greater range of motion when compared with a retroversion of less than 20°. Within these studies, onlay-type RTSA designs were associated with greater forward elevation but less external rotation when compared to inlay-type designs. However, none of the differences in outcome scores and range of motion between the humeral version groups were statistically significant.

## 1. Introduction

The performance of reverse total shoulder replacement (RTSA) for several indications has been steadily increasing over a number of years [1,2,3,4,5,6,7,8]. Since the advent of the Grammont-style prostheses in the 1980s [9], the evolution of RTSA implants has been well-documented, particularly with respect to the glenoid component, with the progressive lateralisation and inferior shift of the glenosphere aiming to reduce scapula notching and component loosening while increasing range of motion (ROM) [10]. In conjunction with these changes, design modifications to the humeral component have also been enacted; for example, the articulating humeral tray was changed from the Grammont-style inlay configuration to an onlay-tray system resulting in increased humeral off-set with the aim of further improving range of motion and reducing implant impingement and scapula notching [11].

An area of continued debate concerns the optimal version in which the humeral stem is implanted. The original designs included neutral retroversion, with a transition to increasing retroversion to allow for more ER. There is a wide variation in practice as to the degrees of retroversion recreated, with biomechanical studies suggesting a compromise of avoiding scapular neck impingement by substituting a more anatomical version compared to increasing the range of motion, specifically external rotation, by implanting the stem in progressive degrees of retroversion [12,13]. Few clinical studies exist that have explicitly examined the effects of altering humeral stem version in RTSA on ROM and functional scores, with the current literature unable to show a distinct advantage of one particular version over another and no clear consensus from expert opinion [14,15,16].

The aim of this study was to systematically review the results of studies describing differently implanted humeral stem versions and humeral tray types (inlay vs. onlay) in RTSA and to answer the following questions:Which humeral stem version gives the best functional outcomes and ROM in RTSA?Which type of humeral design (onlay vs. inlay) within this setting gives the best functional outcomes and ROM?

## 2. Methods

### 2.1. Search Strategy and Criteria

We conducted this systematic review in accordance with the Preferred Reporting Items for Systematic Reviews and Meta-Analyses (PRISMA) statement [17]. The literature was systematically searched for appropriate studies through MEDLINE, PubMed, and Embase databases, and the Cochrane Library, with the final search performed on 31 January 2020. The search was limited to Level I-IV evidence studies on human subjects. A combination of Medical Subject Headings (MeSH) was used as part of the search strategy using the following terms: “reverse”, “total shoulder arthroplasty”, “total shoulder replacement”, “humerus stem”, and “stem version”. In addition, we examined the references in all selected articles to identify studies that had been missed during the initial search.

Two independent reviewers screened all articles eligible for inclusion (SJ, EE). Our inclusion criteria were as follows: a primary RTSA; a minimum two-year follow-up; an explicit description of the implanted humeral stem version in the surgical technique; outcome measures that included ROM, along with absolute Constant scores (CSs) [18] and/or American Shoulder and Elbow Surgeons (ASES) scores [19]. Active ROM was measured in degrees and included forward elevation, abduction, external rotation at 0° shoulder abduction (ER1) and at 90° abduction (ER2), and internal rotation, expressed as a point score for the equivalent vertebral level, as per the Constant score [18]. We excluded the following: ambiguous, or no mention of, the implanted humeral stem version; RTSA for tumour or infection; any revision arthroplasty operation; no availability of an English translation of the study; review articles; biomechanical/simulation studies; and level V evidence.

### 2.2. Data Collection

Studies meeting these criteria were reviewed, and the following data were extracted: study design, number of patients, mean follow-up length, implant used, implanted stem version, humeral tray design, pre- and post-operative range of motion, and Constant score and/or ASES score.

We established three main groups to analyse and compare:Group 1: RTSA with the humeral stem implanted in less than 20° of retroversion;Group 2: RTSA with the humeral stem implanted in 20° retroversion;Group 3: RTSA with the humeral stem implanted in greater than 20° retroversion.

Analysis was performed based on (a) the implanted humeral stem version, and (b) the implant design, i.e., an onlay or inlay system. We sub-divided any studies which had different cohorts of humeral stems that had been implanted with varying versions. In studies that had multiple implants, we did not perform a sub-analysis of the inlay vs. onlay group, if specific data for the prosthesis was not available.

### 2.3. Statistical Analysis

For the functional scores and range of motion, we presented average-weighted means and standard deviations of the study, calculated on the values given by each study and multiplied by the number of patients in each study, divided by the number of all studies in that group to yield the presented figures. The analysis was performed using SPSS (version 27.1, IBM, New York, NY, USA). As we were unable to confirm if all data followed a normal distribution, we used non-parametric testing (Mann–Whitney U test) to compare means.

## 3. Results

### 3.1. Study Characteristics

Our search strategy generated a total of 3572 studies, including those found from citation tracking. Of these, 3482 were removed because of failure to meet the criteria based on the title or because of duplication. The abstracts of the remaining 90 studies were reviewed, resulting in 62 further exclusions. Review of the full-text articles of the remaining 28 studies resulted in an additional 14 studies being excluded and a final inclusion of 14 studies with a total of 1221 shoulders investigated (Figure 1).

Eight of the studies presented Level IV evidence, and six qualified as Level III. All but one of the papers were retrospective studies. The summary of all the included studies is presented in Table 1. The demographics are in Table 1. Overall patient demographics of the three groups are described in Table 2.

#### 3.1.1. Group 1: Humeral Version at Less than 20° Retroversion

Seven studies with a total of 380 patients reported on RTSA where the humeral stem was implanted at less than 20° of retroversion [14,15,16,24,27,28,30]. Of these studies, four were evidence level III, and three were level IV. Mean post-operative ASES scores were reported in three of the seven studies and ranged from 72 to 78.2 with an overall mean of 74.3. Mean post-operative CSs were reported in four of the seven studies and ranged from 56.7 to 69.3, with an overall mean of 59.6. Mean post-operative forward elevation for the studies was 127.9°, mean external rotation (0° abduction, ER1) was 33.6°, mean external rotation (90° shoulder abduction, ER2) was 59.3°, and mean internal rotation was 7.99 points. Five patients had the complication of instability (1.3%).

#### 3.1.2. Group 2: Humeral Version at 20° Retroversion

Five studies with a total of 375 patients reported on RTSA where the humeral stem was implanted at 20° of retroversion [15,16,20,23,29]. Of these studies, four were evidence level III, and one was level IV. Mean post-operative ASES scores were reported in two of the five studies and ranged from 73 to 78.6, with an overall mean of 77.9. Mean post-operative CSs were reported in four of the five studies and ranged from 64 to 73, with an overall mean of 70.1.

Mean post-operative forward elevation for the studies was 129.7°, mean ER1 was 36.4°, mean ER2 was 73.4°, and mean internal rotation was 7.78 points. Three patients had the complication of instability (0.8%).

#### 3.1.3. Group 3: Humeral Version at Greater than 20° Retroversion

Six studies with a total of 466 patients reported on RTSA where the humeral stem was implanted at greater than 20° of retroversion [14,15,21,22,25,26]. Of these studies, two were evidence level III, and four were level IV. Mean post-operative ASES scores were reported in all six studies and ranged from 62 to 81.9, with an overall mean of 73.9. None of the studies reported a post-operative CS. Mean post-operative forward elevation for the studies was 129.1°, mean ER1 was 41.3°, mean ER2 was 61.0°, and mean internal rotation was 7.68 points. Six patients had the complication of instability (1.3%).

### 3.2. Comparison between Groups—Humeral Retroversion

Patients with a humeral stem implanted at 20° of retroversion had the highest post-operative ASES and CS compared with the other two groups (Table 3, Table 4 and Table 5). There were similar post-operative scores between those with either more or less than 20° of retroversion.

Patients with a humeral stem implanted at 20° of retroversion or greater exhibited more forward elevation, but this was not statistically significant. Mean external rotation with the arm in 0° shoulder abduction (ER1) progressively increased as the humeral stem was implanted in greater retroversion, although this was not statistically significant; with the shoulder abducted to 90°, external rotation was greatest at 20° of retroversion compared to other implanted versions, but this was not statistically significant. Mean internal rotation progressively decreased as the humeral stem was implanted in greater retroversion, although this was not statistically significant.

### 3.3. Comparison between Groups—Humeral Stem Designs (Onlay vs. Inlay)

Patients with an onlay-type RTSA demonstrated higher mean post-operative ASES and CS compared to those with an inlay-type RTSA, but this was not statistically significant (Table 6). Those with an onlay-type implant also had more forward elevation (132° vs. 127°) and ER1 (39.5° vs. 34.4°), but less ER2 (50° vs. 65.2°) compared to inlay-type implants; none of these were statistically significant. An onlay-type implant showed a statistically significantly increased internal rotation score compared to the inlay-type implant (9.8 vs. 7.1 points, *p* = 0.048) (Table 7).

## 4. Discussion

The aim of this systematic review was to evaluate post-operative outcomes and ROM of RTSA based on differences in implanted humeral version and humeral stem design. All studies were either level III or IV studies. Across the studies, it has been shown that the implantation of the humeral stem at 20° of retroversion seems to be associated with higher post-operative functional scores. Increased ROM is seen with progressively increased degrees of humeral retroversion. Onlay-type RTSA designs were shown to yield higher post-operative scores, greater forward elevation, greater external rotation with 0° shoulder abduction, and a statistically significant increase in internal rotation compared to inlay-type designs. Inlay RTSA designs were associated with greater degrees of external rotation with the shoulder abducted 90°.

Initial recommendations for implanting the humeral stem in RTSA between 0 and 30° of retroversion were based on expert opinion alone rather than anatomical or clinical studies [31]. With a better understanding of implant design, numerous biomechanical studies have investigated the effects of increasing the retroversion of the humeral stem with respect to stability and range of external rotation in different shoulder positions [32,33]. Favre et al. [34] demonstrated that instability was more common in external rotation, both in neutral and 90° shoulder abduction, with progressive retroversion of the humeral stem. We did not see any significant differences in the rate of dislocation based on humeral stem version. Lädermann et al. [35] demonstrated that external rotation with 0° shoulder abduction was greater in onlay-type humeral stem designs implanted at 20° compared to the traditional inlay-type Grammont humeral-stem designs; however, this was not seen to be the case with shoulder abduction to 90°.

There are few clinical studies that have explicitly studied the influence of humeral stem retroversion on shoulder function after RTSA. Rhee et al. [16] were unable to demonstrate a statistically significant difference in ROM or patient-reported outcome measures (PROMS) between two retrospective patient groups having component retroversion of either 0 or 20°. They did, however, note that performing certain activities requiring greater internal rotation, such as back-washing and bra-fastening, was less problematic when retroversion was 0°. Aleem et al. [14] compared groups with either 10° or less retroversion with groups implanted at 20° or greater retroversion. Both groups demonstrated improvements in ROM and PROMS at two years, but the study was unable to show a difference between these retroversion groups. An issue that was not explicitly described in these papers was that of the residual rotator cuff function, particularly of the infraspinatus and teres minor muscles. While it has been previously suggested that the lack of a residual rotator cuff can negatively affect the outcomes of RTSA [36,37], more recent studies have suggested a diminishing benefit of a residual rotator cuff for shoulder ROM and function [38,39]. In addition, several studies have also described other factors that may influence the functional outcome scores and ROM post-operatively, such as humeral-head inclination angle, glenosphere size, and lateralisation of the stem [40,41]. The variety in implants used between the studies and the likely variation in residual rotator cuff function suggest that the combination of several of these factors may have an ultimate cumulative effect on ROM and function.

One reason for an absence of a difference in the pooled results of our review may be explained by the fact that the predominant PROMS used, the CS and ASES scores, may lack the sensitivity to detect differences in ROM based on the humeral version alone. The compromise in ROM with increasing stem retroversion, as greater external rotation leads to less internal rotation, may cancel out the benefit of one over the other, thus failing to show a clear benefit with a given humeral stem version. Therefore, implantation of the humeral stem at 20° of retroversion, given the higher CS and ASES scores, may be the optimal compromise.

Conversely, Oh et al. [15] demonstrated that patients with more individualised humeral stem implantations had better functional outcome scores and ROM compared with patient groups with fixed 20° humeral stem implantation and hypothesised that this was due to the restoration of the native anatomy, to which the patient had become used to over many years.

With the evolution from Grammont-design prostheses to onlay designs, scapular notching became less common [11]. Biomechanical studies also showed that onlay-stem designs lead to a significant increase in humeral off-set compared with the inlay design and that the tray design influenced humeral off-set more than differences in humeral inclination [42]. Other simulation studies have revealed a correlation between increased humeral off-set and abduction, but less in ER with the arm at the side [43]. Our results revealed that patients with an onlay-type RTSA demonstrated higher mean ASES and Constant scores and higher-mean active-forward elevation, ER1, and internal rotation.

The absence of higher-quality research within this area, in addition to the significant heterogeneity of the examined studies, once again highlights the limitations of systematic reviews drawing more significant conclusions in the absence of level I and II studies. Another limitation of this study was the exclusion of numerous studies that had not explicitly stated the implanted version. While this criterion has allowed us to perform a more precise analysis, for example, excluding studies stating, “the recommended humeral version was used” or those providing a range of implanted versions, it may have altered our results significantly from those we may have obtained if all true stem versions were known due to the vastly increased number of patients that would have been available for analysis. In addition, across the studies, there were a variety of indications for performing RTSA, such as post-traumatic and cuff arthropathy. It is known that outcomes for post-traumatic RTSA differ from those for rotator cuff arthropathy RTSA [44]. The outcomes may have been altered as the PROMS and ROM were not sensitive enough to differentiate between the aetiologies for performing the RTSA. Another limitation with respect to the influence of tray design is the fact that differences between the two design groups were not controlled for confounding factors, such as other design measures or aetiology for RTSA. This systematic review also did not take into account other design parameters, such as humeral inclination, or tray positioning (concentric vs. eccentric), that are known to significantly influence humeral off-set and ROM [35]. We also did not consider differences in the glenoid component.

In conclusion, from available studies, implanting the humeral stem of an RTSA at 20° of retroversion and above appears to produce greater forward elevation and ER1 compared to more anatomical stem versions, albeit with a compromise in internal rotation and external rotation with 90° shoulder abduction. Onlay RTSA designs tend to demonstrate greater post-operative scores, forward elevation, ER1, and internal rotation compared to inlay designs. Inlay designs show greater post-operative external rotation in 90° shoulder abduction. However, given that there were no statistically significant differences in the majority of these scores and measurements, we would recommend Level I and II evidence studies comparing RTSA humeral stems implanted in different degrees of retroversion to confirm if a true difference exists.

## Figures and Tables

**Figure 1 jcm-10-05745-f001:**
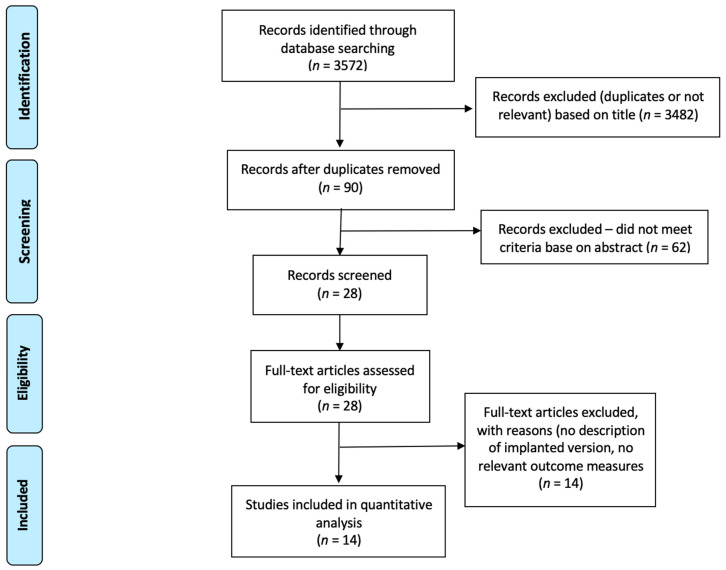
PRISMA flow diagram outlining study selection.

**Table 1 jcm-10-05745-t001:** Summary of all included studies.

Author	Year	Patients	Mean Follow-Up (Months)	Type of Implant	Implanted Stem Version (° Retroversion)	Outcome Measures	Study Details
Aleem et al. [14]	2017	64	24	Inlay	<10°R/>20°R	ASES, ROM	Retrospective study comparing outcomes in two groups of primary RTSR patients based on having their humeral stem implanted in either <10° or >20° of retroversion.
Boileau et al. [20]	2018	38	36	Inlay	20°R	Constant, SSV, ROM	Retrospective study assessing outcomes of patients undergoing primary RTSR for proximal humerus fracture with reattachment of the tuberosities.
Frankle et al. [21]	2005	60	33	Inlay	30°R	ASES, ROM	Retrospective study assessing outcomes of patients undergoing primary RTSR for glenohumeral arthritis associated with severe rotator cuff deficiencies.
Harmsen et al. [22]	2017	232	26.4	Inlay	30°R	ASES, SANE, ROM	Retrospective study assessing outcomes of patients undergoing primary RTSR with a diaphyseal press-fit humeral stem.
Kim et al. [23]	2019	77	70.6	Inlay	20°R	Constant, UCLA, ROM	Retrospective study assessing outcomes of patients undergoing primary RTSR for cuff tear arthropathy and/or a massive irreparable cuff tear.
Leathers et al. [24]	2018	82	40.8/37.2	Inlay	10°R	ASES, ROM	Retrospective study comparing outcomes in two groups of primary RTSR patients, either aged 70 years and older or 65 years and younger.
Oh et al. [15]	2019	80	31.4	Onlay	<20°R/20°R/>20°R	ASES, SST, ROM	Retrospective study comparing outcomes in two groups of primary RTSR patients based on having their humeral stem implanted in either 20° of retroversion or with an individualised native version. Secondary outcomes assessed the effect of subscapularis tendon repair.
Rhee et al. [16]	2015	62	43.3/38.4	Inlay	0°/20°R	Constant, UCLA, ROM, VAS	Retrospective study comparing outcomes in two groups of primary RTSR patients based on having their humeral stem implanted in either 0° or 20° of retroversion.
Samuelsen et al. [25]	2016	67	36	Mixed	27°R	ASES, SST, ROM	Retrospective study assessing outcomes of patients aged 65 years and under, undergoing primary RTSR.
Statz et al. [26]	2016	41	38.4	Mixed	>20°R	ASES, ROM	Retrospective study assessing outcomes of morbidly obese patients undergoing primary RTSR.
Theivendran et al. [27]	2016	124	32	Inlay	<20°R	Constant, OSS, ROM	Retrospective study assessing outcomes of patients undergoing primary RTSR with a trabecular metal glenoid base plate.
Valenti et al. [28]	2011	76	44	Onlay	<20°R	Constant, ROM	Retrospective study assessing outcomes of patients undergoing primary RTSR with a lateralised glenosphere.
Vourazeris et al. [29]	2017	202	39.6/37.2	Onlay	20°R	ASES, Constant, UCLA, SST, ROM	Retrospective study comparing outcomes in two groups of primary RTSR patients based on having either a subscapularis repair or tenotomy.
Young et al. [30]	2011	16	45.6	Inlay	<10°R	Constant, ROM	Retrospective study assessing outcomes of patients with rheumatoid arthritis undergoing primary RTSR.

**Table 2 jcm-10-05745-t002:** Overview of demographics of three groups.

	Group 1 (<20°R)	Group 2 (20°R)	Group 3 (>20°R)
Number of studies	7	5	6
Number of patients	380	375	466
Mean follow-up (months)	35.9	44.8	31.5

**Table 3 jcm-10-05745-t003:** Comparison of Groups 1 and 2.

	<20°R	20°R	*p* Value
**Number of patients**	380	375	
**ASES** **(Mean (Standard Deviation))**	76.0 (3.47)	75.8 (3.96)	0.956
**Constant Score** **(Mean (Standard Deviation))**	62.5 (5.71)	68.1 (3.73)	0.153
**Dislocations (*n*)**	5	3	0.725
**Forward elevation (°)** **(Mean (Standard Deviation))**	127.9 (12.66)	129.7 (6.10)	0.777
**External Rotation (ER1) (°)** **(Mean (Standard Deviation))**	33.6 (10.34)	36.4 (11.41)	0.659
**External Rotation (ER2) (°) (Mean (Standard Deviation))**	59.3 (13.15)	73.4 (1.27)	0.228
**Internal Rotation (0° shoulder abduction) (Points) (Mean (Standard Deviation))**	7.99 (1.41)	7.78 (3.14)	0.893

**Table 4 jcm-10-05745-t004:** Comparison of Groups 1 and 3.

	<20°R	>20°R	*p* Value
**Number of patients**	380	466	
**ASES** **(Mean (Standard Deviation))**	76.0 (3.47)	73.3 (7.55)	0.586
**Constant Score** **(Mean (Standard Deviation))**	62.5 (5.71)	N/A	-
**Dislocations (*n*)**	5	6	0.601
**Forward elevation (°)** **(Mean (Standard Deviation))**	127.9 (12.66)	129.1 (15.81)	0.881
**External Rotation (ER1) (°)** **(Mean (Standard Deviation))**	33.6 (10.34)	41.3 (9.46)	0.192
**External Rotation (ER2) (°) (Mean (Standard Deviation))**	59.3 (13.15)	61.0 (4.24)	0.877
**Internal Rotation (0° shoulder abduction) (Points) (Mean (Standard Deviation))**	7.99 (1.41)	7.68 (4.27)	0.877

**Table 5 jcm-10-05745-t005:** Comparison of Groups 2 and 3.

	20°R	>20°R	*p* Value
**Number of patients**	375	466	
**ASES** **(Mean (Standard Deviation))**	75.8 (3.96)	73.3 (7.55)	0.682
**Constant Score** **(Mean (Standard Deviation))**	68.1 (3.73)	N/A	-
**Dislocations (*n*)**	3	6	0.738
**Forward elevation (°)** **(Mean (Standard Deviation))**	129.7 (6.10)	129.1 (15.81)	0.945
**External Rotation (ER1) (°)** **(Mean (Standard Deviation))**	36.4 (11.41)	41.3 (9.46)	0.462
**External Rotation (ER2) (°) (Mean (Standard Deviation))**	73.4 (1.27)	61.0 (4.24)	0.058
**Internal Rotation (0° shoulder abduction) (Points) (Mean (Standard Deviation))**	7.78 (3.14)	7.68 (4.27)	0.976

**Table 6 jcm-10-05745-t006:** Comparison of inlay and onlay humeral designs.

	Inlay Design	Onlay Design	*p* Value
**Number of patients**	755	358	
**ASES** **(Mean (Standard Deviation))**	74.9 (4.66)	77.9 (3.68)	0.332
**Constant Score** **(Mean (Standard Deviation))**	65.0 (4.53)	66.0 (9.90)	0.842
**Forward elevation (°)** **(Mean (Standard Deviation))**	127.0 (12.88)	132.0 (9.58)	0.459
**External Rotation (ER1) (°)** **(Mean (Standard Deviation))**	34.4 (9.54)	39.5 (12.41)	0.382
**External Rotation (ER2) (°)** **(Mean (Standard Deviation))**	65.2 (10.11)	50 (8.24)	0.210
**Internal Rotation (0° shoulder abduction) (Points) (Mean (Standard Deviation))**	7.1 (1.44)	9.8 (2.33)	0.048

**Table 7 jcm-10-05745-t007:** Comparison of all three groups subdivided by humeral stem design.

	Inlay Design			Onlay Design		
	<20°R	20°R	>20°R	<20°R	20°R	>20°R
**Number of patients (studies)**	283 (5)	145 (3)	327 (3)	97 (2)	230 (2)	31 (1)
**ASES** **(Mean (Standard Deviation))**	73.5 (2.56)	N/A	76 (3.73)	78.2 (17.7)	77.9 (1.83)	81.9 (4)
**Constant Score** **(Mean (Standard Deviation))**	59.8 (5.13)	66.5 (1.54)	N/A	59 (1.58)	73 (N/A)	N/A
**Forward elevation (°)** **(Mean (Standard Deviation))**	118 (11.42)	134 (3.35)	122 (10.26)	129.6 (6.79)	122 (2.59)	141.6 (3.86)
**External Rotation (ER1) (°)** **(Mean (Standard Deviation))**	29.7 (6.82)	41.1 (10.21)	33.6 (3.59)	31.9 (3.68)	26.7 (4.55)	54.5 (3.87)
**External Rotation (ER2) (°)** **(Mean (Standard Deviation))**	65.3 (10.02)	73 (0.81)	58.8 (2.03)	50 (2.24)	N/A	N/A
**Internal Rotation (0° shoulder abduction) (Points) (Mean (Standard Deviation))**	8.86 (2)	6.8 (2.60)	N/A	7.31 (1.32)	12.1 (2.7)	10.7 (2.3)

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
