# Peer review of "Does Humeral Component Version Affect Range of Motion and Clinical Outcomes in Reverse Total Shoulder Arthroplasty? A Systematic Review"

_jcm, 2021, doi:10.3390/jcm10245745_

Round 1

Reviewer 1 Report

Thank you for allowing me to participate in this review. This paper is reviewed on the relationship between humeral component of reverse shoulder arthroplasty (RSA) and clinical results. In addition, the common indication for RSA is the cuff tear arthropathy with irreparable rotator cuff tears. Therefore, humeral retroversion is thought to be influenced on the ROM and clinical results. Thus, this paper potentially has a good impact for readers. Overall, I am in support of this paper, however, would like to see this answer questioned.

The authors did not discuss on residual rotator cuff muscles, especially the teres minor and infraspinatus, although they concern about the subscapularis repair. The atrophy of teres minor with severe the infraspinatus fatty infiltration affected to clinical results and ER strength after RSA. (Boileau P, et al. JSES, 2006) In addition, the shoulder after RSA is also needed to be rotated externally. (Matsuki K, et al. Clin Biomech, 2018) Thus, the retroversion less than anatomical humeral head may support to the deficiency of the external rotational function. However, the conclusion of this review is opposite to the recommended retroversion angle for RSA. I consider this difference can be occurred by the difference of residual rotator cuff. Therefore, I would like authors to review and discuss on the residual rotator cuff, especially the teres minor and infraspinatus for discussing on the retroversion of humeral component.

Author Response

Thank you for reviewing our manuscript. We appreciate the comments of all the reviewers and have subsequently revised our manuscript based on their suggestions.

We have responded to the Reviewer's comments regarding the role of the residual rotator cuff in Lines 293-297.

Reviewer 2 Report

Optimal humeral retroversion is definitely an ongoing debate in reverse shoulder arthroplasty. The present systematic review has the benefit to try to analyze current knowledge regarding this specific issue. It is well written and easy to read.

The study has however some major limitations of which some might be addressed with major revisions:

1. The authors looked at 2 years post-operative outcomes and did not report improvement in clinical scores and ROM between pre and post-op. It would be interesting to look at improvement/loss of ROM (especially in internal and external rotation after RSA) which is a limitation of current RSA designs.

2. The authors stratified studies in three groups which seems reasonable. When looking at both clinical scores and ROM tables it appears however that no single element reached statistically significance except internal rotation with the onlay design. When looking critically, the two degrees of internal rotation seem further questionable to be clinically significant.

3. Table 7: there is data missing regarding internal rotation

4. ROM is definitely influenced by other factors than humeral retroversion and onlay vs. inlay design (eg. glenoid position/lateralization, neck shaft angle, glenosphere diameter).

5. Different glenoid/glenosphere components, glenoid positioning and stem designs used in the different included studies should be reported as they are major limitations of the study and might misslead readers interpretation of the reported results.

6. It would be interesting to report complications between groups, as biomechanical data reported higher risk of dislocation/instability with higher degrees of retroversion. 

7. In my opinion, the benefit of this review article is to highlight that we are currently unaware of the optimal RSA design including humeral retroversion. The discussion and conclusion should therefore clearly highlight that no statistical difference could be found and that while an onlay design might be statistical beneficial for improved internal rotation the benefit of two degrees are clinically questionable. 

8. The absence of significant difference regarding ROM and clinical outcomes between groups of humeral retroversion should be clearly stated in the abstract.

Other limitations which are mentioned by authors seem difficult to address due to the low level of evidence of the published research, therefore limiting conclusions.

1. There is a lack of level I and II studies. The conclusions of a systematic review of level III and IV articles remains therefore by definition weak.

2. Active ROM is not solely related to impingement free range of motion but also to deltoid muscle recruitment and remaining rotator cuff activation. These factors are definitely influenced by NSA, and COR Lateralization, therefore limiting comparability between the reported groups as these factors where not taken into account.

3. Finally the major take home message is that there is a lack of clinical evidence regarding optimal RSA design including humeral retroversion. Due to the variability of prosthetic designs on both the humeral and glenoid side, a comprehensive and none biased comparison of outcomes (clinical score and/or ROM) between different degrees of humeral stem retroversion seems difficult (impossible?) with the currently available papers.

Author Response

1. The authors looked at 2 years post-operative outcomes and did not report improvement in clinical scores and ROM between pre and post-op. It would be interesting to look at improvement/loss of ROM (especially in internal and external rotation after RSA) which is a limitation of current RSA designs.

Response: There were significant absences in the reporting of pre-operative scores and ROM. In the papers, only six of the studies reported pre-operative ASES scores, four reported pre-operative Constant scores and each of the studies had variability in reporting the pre-operative ROM (active elevation in 11 studies, external rotation (neutral shoulder abduction) 12 studies, external rotation (90degrees shoulder abduction) 6 studies, internal rotation 1 study). Therefore, comparisons based on gains of ROM were of limited value given the distribution across the groups and the decision was made to analyse post-operative scores only.

2. The authors stratified studies in three groups which seems reasonable. When looking at both clinical scores and ROM tables it appears however that no single element reached statistically significance except internal rotation with the onlay design. When looking critically, the two degrees of internal rotation seem further questionable to be clinically significant.

Response: This is now covered in the discussions section (line 345 onwards)

3. Table 7: there is data missing regarding internal rotation

Response: This has now been completed.

4. ROM is definitely influenced by other factors than humeral retroversion and onlay vs. inlay design (eg. glenoid position/lateralization, neck shaft angle, glenosphere diameter).

Response:  This is now covered in the discussions section (line 286 onwards)

5. Different glenoid/glenosphere components, glenoid positioning and stem designs used in the different included studies should be reported as they are major limitations of the study and might misslead readers interpretation of the reported results.

Response: This is now covered in the discussions section (line 286 onwards)

6. It would be interesting to report complications between groups, as biomechanical data reported higher risk of dislocation/instability with higher degrees of retroversion. 

Response:  This is now covered in the results section at the end of each of the group’s paragraphs and in the comparison tables.

7. In my opinion, the benefit of this review article is to highlight that we are currently unaware of the optimal RSA design including humeral retroversion. The discussion and conclusion should therefore clearly highlight that no statistical difference could be found and that while an onlay design might be statistical beneficial for improved internal rotation the benefit of two degrees are clinically questionable. 

Response: This is now covered in the discussions section (line 345 onwards)

8. The absence of significant difference regarding ROM and clinical outcomes between groups of humeral retroversion should be clearly stated in the abstract.

Response:  This is now covered in the abstract (line 68 onwards)

Round 2

Reviewer 1 Report

Thank you for allowing me to participate in this review again. Authors added their answer for may question as the limitation. 

Author Response

Thank you.